# Development and Validation of a Rating Scale of Pain Expression during Childbirth (ESVADOPA)

**DOI:** 10.3390/ijerph17165826

**Published:** 2020-08-12

**Authors:** Silvia Navarro-Prado, María Angustias Sánchez-Ojeda, Adelina Martín-Salvador, Trinidad Luque-Vara, Elisabet Fernández-Gómez, Elena Caro-Morán

**Affiliations:** 1Department of Nursing, Faculty of Health Sciences, University of Granada, 52071 Granada, Melilla, Spain; silnado@ugr.es (S.N.-P.); ademartin@ugr.es (A.M.-S.); triluva@ugr.es (T.L.-V.); elisabetfdez@ugr.es (E.F.-G.); 2Department of Physiotherapy, Faculty of Health Sciences, University of Granada, 52071 Granada, Melilla, Spain; ecaro@ugr.es

**Keywords:** scale, pain, validation study, psychometric test, parturient, childbirth

## Abstract

One of the most representative symptoms during childbirth is pain, which is one of the most prominent concerns of pregnant women. There are different instruments to assess pain, all of which require interrupting the woman, thus interfering with the intimacy of childbirth. This study seeks to develop and validate a rating scale of the expression of childbirth pain that does not require the mother’s attention and respects her privacy during labor. The study was conducted at a regional hospital in a border town in southern Spain between November 2018 and September 2019. Scale items were developed following a review of the scientific literature, and experts judged the content validity. After a pilot test, the scale was psychometrically evaluated. The psychometric tests consisted of internal consistency analysis, exploratory factor analysis, and determination of the content, construct, and convergent validity. The scale was evaluated by 36 experts in the field and was then applied to 55 women during the active phase of childbirth. The final version of the Rating Scale of Pain Expression during Childbirth (in Spanish, *Escala de Valoración de la Expresión del Dolor durante el Trabajo de Parto*—ESVADOPA) consists of six items in two dimensions. The scale had a Cronbach’s alpha coefficient of 0.78, and the content validity measured by Aiken’s V co-efficient was also 0.78. The exploratory factor analysis yielded two dimensions that explained 68.08% of the total variance. For convergent validity, a comparison was made with the visual analogue scale, yielding a medium–high value of 0.641. As indicated by the internal consistency and by the content and construct validity outcomes, the ESVADOPA successfully measures pain expression during childbirth and represents a suitable tool for pain expression during birth without the need for intervention or the need for the mother to speak the same language as the midwife.

## 1. Introduction

One of the most representative symptoms during the birthing process is pain [1], which has been described as one of the most severe and exhausting kinds of pain humans experience, including not only its physical dimension but also its psychological dimension [1,2,3]. The pain of childbirth is one of the most prevalent concerns of pregnant women [4]. The perception of it is intimately related to numerous aspects, such as previous experiences, emotional factors, and physical factors, all of which are modulated by cultural and social factors [5,6,7,8]. 

Likewise, the expression of the childbirth pain perceived by women depends on different aspects [9]. The memory of experiences of pain; the healthcare received during earlier painful experiences, whether or not the woman has given birth; and the socio-cultural environment modulate how women express pain during childbirth [4,6,10].

Another factor that affects the expression of pain by women is fear. The ignorance of primiparous women regarding their ability to cope with childbirth leads, in most cases, to the first birth being the most painful, since later the women know about the process, and fear decreases with knowledge [5,11,12]. Women who had a bad experience in a previous birth often have similar pain and anxiety levels as primiparous women, since, although they know what they are facing, the previous negative experience increases their anxiety and fear, leading to the perception and expression of childbirth pain [10,13].

Knowledge can be influenced by maternal education, which can reduce both fear and pain, but for this education to be effective, it is necessary for the midwife and the pregnant woman to speak the same language [14]. With the increase in human migration, the number of deliveries by women who cannot verbally communicate with their obstetrician has increased, causing a decrease in care quality [2,6,13]. 

Regardless of whether the midwife and the parturient woman can communicate with one another verbally, the period of birth (when women are likely to experience fear, uncertainty and pain) is not a favorable time to ask someone to respond to a questionnaire. Additionally, during birth, the rating provided by the subject may vary for a number of reasons, including the influence of the normal progression of birth and fluctuations in mood and sense of security. Other studies have reported that women have expressed a desire not to be disturbed during birth and a preference to be interviewed after childbirth [15].

For this reason, numerous midwives’ groups have demanded the creation of visual material to facilitate the interaction between the obstetrician and the pregnant woman and to help interpret the woman’s symptoms during childbirth, which would give caregivers more knowledge of the pain tolerance and pain level of each woman [5,11]. Such visual materials would contribute to the current trend of respecting the childbirth process. Currently, obstetric care is increasingly focused on reducing interventions and interfering with the mother as little as possible at the time of birth [16]. It is for this reason that it is necessary to create an instrument that follows this trend and does not require interrupting the parturient woman, thus respecting her intimacy and reducing the interruptions.

Scales currently used for pain assessment are based on the perception of the woman, who scores her pain on an analogue scale, a numbered scale, or a color gradient. The pain ranges from an absent to severe to the worst imaginable pain, as in the case of the Visual Analogue Scale (VAS), which facilitates assessing pain intensity. Although the VAS is highly regarded and widely used in clinical research, it requires the birthing woman to complete it. Furthermore, the midwife must explain to the parturient woman how the scale functions, for which it is necessary that they speak the same language [15,16,17]. However, as birth progresses, the birthing woman’s capacity for self-evaluation can vary, and there are occasions when such scales are only appropriate for the beginning of birth. Thus, they are insufficient to evaluate pain occurring close to the expulsion stage of birth. Because they are complex and require a long time to complete, other scales intrude too much on the privacy of the women and are often rejected by parturient women [1,18,19,20].

Given the above, the objective of this study is to develop and validate a rating scale for the expression of childbirth pain that is completed solely on the basis of the midwife’s observation, without interfering in the birthing process or being influenced by a language barrier and which is also low-cost and fast to apply.

## 2. Materials and Methods

### 2.1. Sample

The study was conducted at a regional hospital in a border town in southern Spain between December 2018 and September 2019. The initial sample was 76 parturient women with full-term pregnancies of a single fetus who were evaluated in the active phase of childbirth. The final sample was 55 women in labor, since women were excluded who delivered via caesarean section and/or were administered epidural anesthesia in the early moments of childbirth and thus could not be evaluated because they did not reach the active phase. The guidelines and ethical principles for medical research in human beings established by the World Medical Association in the Declaration of Helsinki, in its latest version from the 64th General Assembly, Fortaleza, Brazil, of October 2013, were followed. The knowledge and approval of the hospital’s ethics committee was ensured. Likewise, the willingness of women to voluntarily participate in the study was respected, preserving their privacy and well-being throughout. All parturient women signed an informed consent form for participation in the study, which detailed its purpose. This article was approved by the ethics committee of Comarcal Hospital of Melilla (Spain) with the registration number 201800500007736 on the 14th of September of 2018.

### 2.2. Development of the Scale

During the months of November and December 2018, a search of the current scientific literature was conducted to guide the development of the scale and its items. In January 2019, the scale was created by the midwife on this research team. The Campbell scale [17] was used as a basis for developing this new measuring instrument that would assess pain in patients who were unable to spontaneously communicate, since that scale fit our objectives of non-interruption of the parturient woman and fast evaluation. The Campbell scale has five items that are scored from 0 to 2: face, restlessness, muscle tone, vocalization, and consolability. The lower the score, the lower the pain expressed by the patient.

Adaptation of the above scale to the Rating Scale of Pain Expression during Childbirth (*Escala de Valoración de la Expresión del Dolor durante el Trabajo de Parto*—ESVADOPA, in Spanish) was performed by considering the care experience of the research team’s midwife, an extensive reading of the current scientific literature, and the contributions of the rest of the members of the research team. The result of this adaptation is a scale that assesses six items during childbirth. These items are *facial muscles*, *body response*, *verbal response*, *restlessness*, *ability to relax*, and *vegetative symptoms.* Each of these items is scored from 0 to 3, with 0 meaning absence of pain expression and 3 meaning maximum pain expression. It is important to consider that low scores mean that the parturient woman does not express pain. However, such scores do not mean that the woman does not feel pain. Therefore, the midwife must pay constant attention. 

To fill in the scale, interrupting or interviewing the woman in labor is not necessary. The midwife only needs to observe the woman’s reaction during one contraction to make the assessment; if any of the items are not clear, assessment is made based on two consecutive contractions. After the evaluation, a score is obtained that is categorized as follows: <1: *Does not express pain*; 1–6: *Expresses mild pain*; 7–12: *Expresses moderate pain*; and 13–18: *Expresses intense pain.*

The bibliography used for the adaptation of each item is listed in Table 1.

### 2.3. Procedure

ESVADOPA was presented to a group of 36 experts, who were selected for convenience and intentionality. Ten of the experts were academic doctors from the University of Granada or the University of Jaén (Spain), and all of the experts had extensive experience in obstetrics and pain assessment because of their care, academic, and/or research experience. These experts were 24 midwives, five gynecologists, three nurses, two psychologists, one anesthetist, and one neurologist.

The evaluation was performed in two sessions one week apart. First, the objectives of both the scale and of the study were presented and explained to each of the evaluators. In this session, they were asked to make suggestions they considered appropriate. In the second phase, group sessions were organized for sharing suggestions. After incorporating the suggestions that were considered appropriate in both sessions, the 36 experts concluded that the scale could measure the expression of childbirth pain.

The resulting modified scale was then applied to a group of 55 women seen during childbirth. To select the sample, simple random sampling was performed using data provided by the hospital administration, which was used to calculate the average number of births for Spanish and migrant women. Once the mean was obtained for births in the hospital between 2014 and 2018, the sample size necessary for a confidence level of 95.45% was calculated, which provided a K coefficient equal to two and a required sample size of 55. The mean age was 28.45 years (SD ± 6.9). A total of 57% of the parturient women experienced a language barrier, whereas 43% experienced no difficulty in linguistic communication. The evaluation was performed during the active phase of labor at 7–10 cm dilation and with the absence of a desire to push. The evaluation was performed during the active phase of birth in women who either rejected epidural anesthesia or could not have it administered for different obstetric/medical reasons. The evaluating midwife evaluated each woman during one contraction to assess the progression of pain expression as the contraction progressed through its increment, acme, and decrement. After the disappearance of this contraction and after allowing the woman to recover, they were asked to assess the pain they had experienced during this contraction on the visual analogue scale (VAS). The evaluation by the evaluating midwife and by the parturient woman had to be performed based on the same contraction so that the progression of childbirth would not alter the correlation (if any) between the two evaluations due to the increased intensity of contractions inherent to normal labor progression. In case the evaluating midwife needed to assess two consecutive contractions due to doubts, the VAS was applied after the end of the second contraction. This information was always collected by the same person in the research group to avoid bias between evaluators.

### 2.4. Data Analysis

In the psychometric analysis of ESVADOPA, the following descriptive statistics were calculated: Mean, standard deviation, and all other data provided by IBM Statistical Package for the Social Sciences (SPSS) version 25.0 software package, according to the descriptors to be analyzed.

The reliability (internal consistency) of the scale was tested using Cronbach’s alpha coefficient and the total correlation of each item with the scale score. Alpha values equal to or greater than 0.70 are considered adequate [29].

The participation of the 36 experts allowed us to calculate the content validity of ESVADOPA using Aiken’s V coefficient in Excel. The value of Aiken’s V varies between 0 and 1, and values lower than 0.70 cause rejection of the null hypothesis [30].

To determine the construct validity, we analyzed the factor structure using principal component exploratory factor analysis (EFA). The adequacy of this analysis was verified using the Kaiser–Meyer–Olkin (KMO) test and the Bartlett sphericity test [31]. The number of factors for extraction was based on the KMO eigenvalue criterion (eigenvalue ≥ 1) and the evaluation of the scree plot.

Convergent or concurrent validity was evaluated using the Pearson correlation coefficient, measuring the association between the ESVADOPA and VAS scales [17] (gold-standard reference test) (Table 2).

## 3. Results

### 3.1. Reliability

Cronbach’s alpha coefficient is a model of internal consistency based on the mean of the correlations between the items. Its advantages include the possibility of evaluating how much the reliability of the scale would improve (or worsen) if a certain item were excluded. In this study, the Cronbach’s alpha coefficient was 0.784, which indicated acceptable internal consistency for a scale with only six items (Table 2).

The corrected total correlation of elements is called the homogeneity index (HIc). If the HIc of an item is ≤0, the item is removed [31]. The item–total correlation coefficients ranged from 0.178 to 0.681, with the item Vegetative Symptoms (VS) having the lowest correlation and the item Verbal Response (VR) having the highest correlation (Table 2).

Likewise, from the analysis of the reliability of the scale, an intraclass correlation coefficient (ICC) = 0.784 was obtained, which is considered acceptable [32].

### 3.2. Validity

Through content validity, we tried to determine how representative the scale items were of the criterion to be evaluated (childbirth pain). In our case, we had 36 expert opinions. To calculate the content validity, we used Aiken’s V coefficient, and the value obtained from the scale was 0.78, which is considered acceptable. To arrive at this value, a spreadsheet constructed for this purpose by Cordón [33] was used, following the indications previous studies [30,34]. 

A principal component EFA with oblimin rotation was performed (considering correlated factors). The KMO sample adequacy measure is an index that compares the magnitude of the observed correlation coefficients with the magnitude of the partial correlation coefficients [35]. If the KMO value is close to 1, the data have an excellent fit to the factorial model. Values below 0.6 are considered mediocre. In our case, the KMO value obtained was 0.796. The Bartlett sphericity test was significant (*p* < 0.000), (chi-squared = 99.474), (df = 15), which confirmed the adequacy of the data for performing a principal component analysis (Table 3). The scree plot was used to determine the best number of factors. It indicated that only the eigenvalues of the first two variables were greater than 1, meaning that these two variables summarized the rest, representing them in a coherent way; that is, these two variables were the two main components that summarized all the information (Figure 1).

The total variance explained showed that the first two components summarized 68.083% of the total variance (Table 4). The value of the determinant was 0.143, which was very small, indicating that the degree of intercorrelation between the variables was very high, an initial condition that the principal component analysis should meet.

According to the results, ESVADOPA consists of two dimensions: the first comprising four items, and the second comprising two items. The research group decided to name the first the basic dimension and the second the complementary dimension.

The convergent or concurrent validity was calculated by comparing the results obtained in ESVADOPA with another, existing method that has been validated and considered the gold standard [36]. In our case, the VAS was used to measure pain intensity. The Pearson correlation coefficient between the two scales was a medium–high 0.641, which was statistically significant [37]. (Table 5).

## 4. Discussion

ESVADOPA is a new scale that has been developed to assess the expression of childbirth pain in such a way that only the midwife’s observation is needed, without interfering in the childbirth process, regardless of the language of the woman, being a low-cost scale with fast application. Due to its ease of use and the number of items included, it is a feasible instrument.

Although there are scales that assess pain, such as the VAS [17], there are few that specifically focus on childbirth pain [1,38] and none that do not require interrupting the woman. For this reason, it was necessary to create and validate a scale that does not involve women in the measurement of their pain in order not to interfere with any part of the childbirth process. The scale has other advantages, including that it is not necessary for the woman and the midwife to speak the same language and that the parturient woman is not asked to complete a scale that she considers inaccurate because she believes that her pain exceeds the scale’s parameters. This scale is not only ideal for the care provided by midwives but also for other professionals who assist during childbirth. This includes anesthesiologists, since the scale improves the estimation of the time to administer epidural anesthesia and may help reduce possible dystocia by its prompt administration. The scale also allows researchers to conduct rigorous studies related to the evaluation, care, and treatment of pain during the birthing process without having to disturb the women. ESVADOPA has been designed after an extensive review of the literature and the incorporation of the changes and suggestions made by numerous experts.

There are multiple psychometric tests for the validation of a health measurement instrument [39,40]. To validate ESVADOPA, the reliability was evaluated using Cronbach’s alpha, the content validity through expert judgement, the convergent validity by comparing two pain measurement instruments, and the construct validity by performing a factorial analysis. All confirmed that the scale created here has reliability and validity equal to or even better than other scales used for pain assessment.

The internal consistency of ESVADOPA was above the established cut-off [40], with Cronbach’s alpha = 0.784, which indicates good reliability, meaning that the scale is a good instrument for measuring the expression of pain during childbirth.

ESVADOPA was revised with basis on the judgement of a broad panel of experts made up of different health professionals with extensive experience in obstetrics and pain assessment, indicating a high content validity of the scale. According to the scientific literature consulted, there is no agreement on the number of experts who should participate in a validation, with some authors arguing for two judges [41] and others advocating approximately 30 [42,43,44]. Because a greater number of experts will lead to more information being collected on the scale [45,46,47,48], our expert panel comprised 36 professionals with extensive healthcare, teaching, and/or research experience related to childbirth pain.

To strengthen the construct validity of the scale, an EFA was conducted, showing stability in two main dimensions out of Facial Muscles, Body Response, Verbal Response, Restlessness, Ability to Relax, and Vegetative Symptoms, in such a way that acceptable internal consistency was achieved. Therefore, ESVADOPA is considered a highly valid scale to measure childbirth pain without the need to disturb the parturient woman.

Lastly, in the convergent validation of the ESVADOPA scale, we examined its correlation with the VAS, a scale widely used to measure pain. The Pearson correlation coefficient between the two scales showed that the degree of association between the two was positive, meaning that ESVADOPA also measures the expression of pain in childbirth, with a medium–high correlation coefficient of *r* = 0.641, as found in other studies [21,36,40,49]. 

The sample of 55 women on whom we validated the scale was considered sufficient given the number of births at the hospital where the study was conducted. Other studies have validated scales with a similar or even smaller numbers of participants [1,50,51].

After all the psychometric analyses of the scale, good reliability and content validity data were obtained, so we argue that this scale is a useful and even ideal tool to evaluate the expression of pain in women during childbirth. To all this, we can add its ease of application, since only the midwife is needed as an evaluator, without having to interfere with the privacy of the parturient woman enable the scale to be used in a broad context. Future research directions may also be highlighted.

## 5. Conclusions

ESVADOPA adequately measures the expression of childbirth pain, as demonstrated by its internal consistency, its content validity, and its construct validity. This scale can be helpful and effective when used by midwives and other healthcare personnel in the care of a birthing mother. The routine application of ESVADOPA may help increase the quality of care at the time of birth.

Limitations: It would have been desirable to compare ESVADOPA with other pain assessment scales to strengthen its concurrent validity, but since there are no scales that do not require interrupting or interviewing the woman, it was decided to use only the VAS, since it is the scale that interrupts the parturient woman the least. It is necessary to complement this quantitative study with a qualitative one in which the opinion of women after birth is taken into account. Future studies in which the scale is used by more than one evaluating midwife will increase its objectivity.

The sample size was low for two main reasons. A single midwife did all evaluations, to avoid inter-evaluator biases, which limited the number of parturient women due to the evaluator’s work shift. Performing the evaluation in the active phase of childbirth limited the sample size even more, since many women were administered epidural anesthesia during the latent phase, or it was decided to terminate the pregnancy by caesarean section before the active phase of childbirth.

## Figures and Tables

**Figure 1 ijerph-17-05826-f001:**
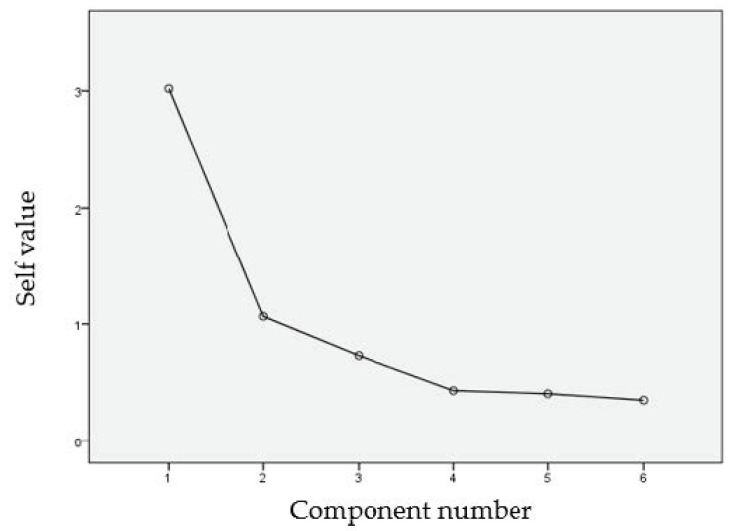
Scree plot.

**Table 1 ijerph-17-05826-t001:** Scientific literature used for the preparation and adaptation of ESVADOPA (in Spanish, *Escala de Valoración de la Expresión del Dolor durante el Trabajo de Parto*).

Item	Sub-Item: Score/Meaning	References	Articles
FACIAL MUSCLES (FM)	0. Relaxed during the entire contraction. 1. Relaxed with slight facial tightening during most of the contraction. 2. Frowning/grimacing/clenched teeth only during the peak of contraction. 3. Frowning/grimacing/clenched teeth during the entire contraction.	[3,19,21,22,23,24,25]	Women’s experiences of pharmacological and non-pharmacological pain relief methods for labor and childbirth: a qualitative systematic review (2019). Moving Beyond the 0–10 Scale for Labor Pain Measurement (2016). Development, psychometric assessment, and predictive validity of the comprehensive breastfeeding knowledge scale (2020). Validación de la Escala de Conductas Indicadoras de Dolor para valorar el dolor en pacientes críticos, no comunicativos y sometidos a ventilación mecánica: resultados del proyecto ESCID (2011). Instruments measuring pregnant women’s expectations of labor and childbirth: A systematic review (2020). Expression of Pain Behaviors and Perceived Partner Responses in Individuals with Chronic Pain: The Mediating Role of Partner Burden and Relationship Quality (2018). Monitorización del dolor. Recomendaciones del Grupo de Trabajo de Analgesia y Sedación de la SEMICYUC (2008).
BODY RESPONSE (BR)	0. Relaxed during the entire contraction. 1. Relaxed with slight contraction of hands, arms, and/or toes and legs during most of the contraction. 2. Increased: flexion of the fingers, arms, and/or toes and legs during the peak of contraction. 3. Increased: flexion of fingers, arms, and/or legs during the entire contraction.	[3,21,23,24,25]	Women’s experiences of pharmacological and non-pharmacological pain relief methods for labour and childbirth: a qualitative systematic review (2019). Development, psychometric assessment, and predictive validity of the comprehensive breastfeeding knowledge scale (2020). Instruments measuring pregnant women’s expectations of labor and childbirth: A systematic review (2020). Expression of Pain Behaviors and Perceived Partner Responses in Individuals with Chronic Pain: The Mediating Role of Partner Burden and Relationship Quality (2018). Monitorización del dolor. Recomendaciones del Grupo de Trabajo de Analgesia y Sedación de la SEMICYUC (2008).
VERBAL RESPONSE (VR)	0. In silence or fluid conversation during the entire contraction. 1. Mild moans and sobs during most of the contraction. 2. Shouts, complains, grunts, and sobs at the peak of contraction. 3. Shouts, complains, grunts, and sobs during the entire contraction.	[3,21,23,24,25]	Women’s experiences of pharmacological and non-pharmacological pain relief methods for labour and childbirth: A qualitative systematic review (2019). Development, psychometric assessment, and predictive validity of the comprehensive breastfeeding knowledge scale (2020). Instruments measuring pregnant women’s expectations of labor and childbirth: A systematic review (2020). Expression of Pain Behaviors and Perceived Partner Responses in Individuals with Chronic Pain: The Mediating Role of Partner Burden and Relationship Quality (2018). Monitorización del dolor. Recomendaciones del Grupo de Trabajo de Analgesia y Sedación de la SEMICYUC (2008).
RESTLESSNESS (R)	0. Calm, relaxed, normal movements during the entire contraction. 1. Calm, relaxed, slight movements indicating restlessness during most of the contraction. 2. Occasional movements indicating restlessness and/or changes in position at the peak of contraction. 3. Continuous movements indicating restlessness and/or changes in position during the entire contraction.	[3,8,21,23,25,26,27]	Women’s experiences of pharmacological and non-pharmacological pain relief methods for labour and childbirth: A qualitative systematic review (2019). Creating a positive perception of childbirth experience: Systematic review and meta-analysis of prenatal and intrapartum interventions (2018). Development, psychometric assessment, and predictive validity of the comprehensive breastfeeding knowledge scale (2020). Instruments measuring pregnant women’s expectations of labor and childbirth: A systematic review (2020). Monitorización del dolor. Recomendaciones del Grupo de Trabajo de Analgesia y Sedación de la SEMICYUC (2008). El alivio del dolor en el parto. Empoderamiento y vulnerabilidad de las mujeres en la toma de decisiones. Estudio cualitativo (2020). Labour pain in women with and without severe fear of childbirth: A population-based, longitudinal study (2018).
ABILITY TO RELAX (AR)	0. Relaxed and calm throughout the contraction. 1. Relaxes with the touch and/or voice of the companion or health professional. 2. Begins to present difficulties with relaxing with the touch and/or voice of the companion or health professional. 3. Rejects the touch and/or the voice of the companion or health professional.	[3,8,21,23,25,26,27]	Women’s experiences of pharmacological and non-pharmacological pain relief methods for labour and childbirth: A qualitative systematic review (2019). Creating a positive perception of childbirth experience: systematic review and meta-analysis of prenatal and intrapartum interventions (2018). Development, psychometric assessment, and predictive validity of the comprehensive breastfeeding knowledge scale (2020). Instruments measuring pregnant women’s expectations of labor and childbirth: A systematic review (2020). Monitorización del dolor. Recomendaciones del Grupo de Trabajo de Analgesia y Sedación de la SEMICYUC (2008). El alivio del dolor en el parto. Empoderamiento y vulnerabilidad de las mujeres en la toma de decisiones. Estudio cualitativo (2020). Labour pain in women with and without severe fear of childbirth: A population-based, longitudinal study (2018).
VEGETATIVE SYMPTOMS (VS)	0. No vegetative symptoms. 1. Sweating and/or nausea. 2. Sweating, nausea, and/or dizziness. 3. Sweating, nausea, vomiting, dizziness, increased blood pressure (BP), tachycardia, and/or dilated pupils.	[5,10,13,23,25,27,28]	Tokophobia (fear of childbirth): Prevalence and risk factors (2018). Higher prevalence of childbirth related fear in foreign born pregnant women—Findings from a community sample in Sweden (2015). Interventions for reducing fear of childbirth: A systematic review and meta-analysis of clinical trials (2018). Instruments measuring pregnant women’s expectations of labor and childbirth: A systematic review (2020). Monitorización del dolor. Recomendaciones del Grupo de Trabajo de Analgesia y Sedación de la SEMICYUC (2008). Labour pain in women with and without severe fear of childbirth: A population-based, longitudinal study (2018). Definitions, measurements and prevalence of fear of childbirth: A systematic review (2018).

**Table 2 ijerph-17-05826-t002:** Descriptive statistics, internal consistency, and matrix of components of the exploratory factor analysis.

	Mean	Standard Deviation	Corrected Total Correlation of Elements (HIc)	Cronbach’s Alpha if the Element Were Eliminated	Component 1	Component 2
FM	2.15	0.621	0.593	0.738	0.257	
BR	2.22	0.599	0.562	0.746	0.296	
VR	1.82	0.819	0.681	0.710	0.293	
R	2.29	0.685	0.660	0.718	0.267	
AR	1.76	0.637	0.564	0.744		0.372
VS	1.00	0.638	0.178	0.828		0.819

FM: Facial Muscles; BR: Body Response; VR: Verbal Response; R: Restlessness; AR: Ability to Relax; VS: Vegetative Symptoms. HIc: Corrected homogeneity index.

**Table 3 ijerph-17-05826-t003:** Confirmatory factor analysis.

KMO and Bartlett’s Test		
KMO Measure of Sampling Adequacy		0.796
Bartlett’s sphericity test	Approx. chi-squared	99.474
	df	15
	Sig.	0.000

**Table 4 ijerph-17-05826-t004:** Total explained variance.

Component	Initial Eigenvalues			Extraction Sums of Squared Loadings			Rotation Sums of Squared Loadings		
	Total	Variance %	Cumulative %	Total	Variance %	Cumulative %	Total	Variance %	Cumulative %
1	3.020	50.330	50.330	3.020	50.330	50.330	2.858	47.634	47.634
2	1.065	17.752	68.083	1.065	17.752	68.083	1.227	20.449	68.083
3	0.730	12.161	80.244						
4	0.431	7.186	87.431						
5	0.405	6.749	94.180						
6	0.349	5.820	100.000						

Extraction method: principal component analysis. Oblimin rotation.

**Table 5 ijerph-17-05826-t005:** Bivariate correlations.

		EVA	ESVADOPA
VAS	Pearson correlation	1	0.641 **
	Sig. (two-tailed)		0.000
ESVADOPA	Pearson correlation	0.641 **	1
	Sig. (two-tailed)	0.000	

** The correlation is significant at the 0.01 level (two-tailed).

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
