# Peer review of "Development and Validation of a Rating Scale of Pain Expression during Childbirth (ESVADOPA)"

_ijerph, 2020, doi:10.3390/ijerph17165826_

Round 1

Reviewer 1 Report

Thank you for the opportunity to review this manuscript. This is a well written paper and well conducted study evaluating a new pain assessment tool for women in labour. I recommend its inclusion in this journal after the following amendments:

Introduction

The introduction presents a comprehensive summary of current conceptions of labour pain, highlighting its complexity. It could be improved by better justifying the need for a pain assessment tool that does not disturb the parturient woman. The authors present the main reason as being that midwives and women may not always speak the same language and this places a barrier to adequate maternal education. To me, the more obvious problem that this would present is difficulty for a woman to complete a pain questionnaire that is not in her native language. And a second issue would be possible difficulty in developing rapport between the woman and her midwife (or other caregiver) due to the language barrier. Furthermore, women themselves have reported that they do not wish to be disturbed with a pain assessment during their labour (see Jones, L. E., Whitburn, L. Y., Davey, M.-A., & Small, R. (2015). Assessment of pain associated with childbirth: Women's perspectives, preferences and solutions. Midwifery, 31(7), 708-712).

The second last paragraph of the introduction (starting on line 63) is critical. I would suggest expanding on the ‘distortion’ that occurs in typical unidimensional scales as labour progresses (Jones et. al may also be useful for this). This paragraph is also critical because it will influence the use of the VAS as the ‘gold standard’ for validation of the ESVADOPA. I feel that the VAS should be discussed in the introduction prior to its first mention on line 154.

Materials and methods

2.2 Development of the scale

Line 109 lists the scoring categories of the scale. I think these categories need further consideration / discussion. The naming of these categories suggests prioritisation of the pain intensity over any other dimension of the woman’s pain. This may be problematic because many women express intense pain (particularly in late labour) however at the same time feel they are coping well / self-managing the pain. In this scenario it would be inappropriate for their caregiver to presume that the intense pain necessitates intervention (non-pharmacological or otherwise). Likewise, a woman may be experiencing ‘mild’ or ‘moderate’ pain, but her affective or evaluative perception of the pain or of her sense of being able to cope means that she actually does need assistance from her caregivers. It is difficult for me to fully appreciate how the authors have come to name these scoring categories without more information about the contents of, and justification for, each of the scale items.

Furthermore, was any feedback sought from the target population (i.e. the women)? This would contribute to the content validity of the scale and could have contributed to the issues raised above about the naming of the score categories. If this was not undertaken it should be included in limitations.

Table 1

I do not feel that this bibliography is very useful at present (utilising 3 columns to list references that contributed to the development of each item in the ESVADOPA). I would perhaps suggest that this table is used as an opportunity to present details of the contents of each of the six items of the scale (with reference to the literature that informed each item, if desired). The manuscript at present does not contain any detail of the contents of each item, some of which are not very intuitive (vegetative symptoms) or which seem to overlap (ability to relax and restlessness).

Furthermore, a number of the years listed in column 4 do not correspond with the year of publication of the article that it is relating to. This should be checked. Also, there is at least one typo (the first reference listed – ‘Wome’s’ should be ‘Women’s’).

2.3 Procedure

This scale was applied to a group of 55 women. How was this sample size determined? Also, what was the characteristics of the included women and, critically, were any women included who did not speak the same language as their midwife?

2.4 Data analysis

Line 146 – What do you mean by ‘possible participation’?

Line 154 – I believe this was the first use of the abbreviation VAS. If so, it should be written in full before abbreviation used.

Discussion

Line 216 – Again, the comment ‘...not necessary for the woman and the midwife to speak the same language..’ made me wonder whether this validation study included women in this category or not. If not, this should be included in the limitations.

Line 217 – ‘..the woman cannot overestimate (over-report) the pain she feels’. I think this is a really sensitive idea and should be re-thought. According to the International Association for the Study of Pain, an individual’s pain experience is subjective and therefore is exactly as they state it is. In the context of labour, a woman who is expressing great distress from her pain should not be dismissed for ‘over-reporting’ her pain, even if the physiological state of her labour suggests that she is progressing normally and/or contractions are not yet at their greatest intensity. Pain is a very complex experience and while the intensity of her pain may not yet have peaked the affective and cognitive dimensions of her pain may be causing her distress and suffering.

Conclusions

Line 267 – Further discussion about the limitations of VAS to assess the validity of this scale should be mentioned.

Other limitations not mentioned include lack of evaluation of inter-rater reliability and lack of feedback from target population to confirm content validity and scale meaningfulness.

Author Response

Line 54: now lines 58-64 have tried to better justify wome´s involvement in pain assessment and their desire not to be disturbed during labor.

Line 67: now lines 74-84, an attempt has been made to better explain the disadvantage of VAS and the word “overestimate” has been deleted as it led to confusion (line 121-123)

Line 109: this suggestion is added in the limitations

Table 1: the items corresponding to each section are added to make it clearer and the years are rectified.

Procedure: more information on the sample of woman studied.

Line 146: now lines 152-154 the text is modified avoiding the word “overestimate” which led to confusion.

Line 216, 217 and 267: modified with the previous contributions.

Reviewer 2 Report

Attached

Author Response

Lines 14-15: the beginning of the sentence is changed.

Line 29: the conclusion is extended as advised.

Line 85: we do not have another Ethics Committee identification number. The scanned document was submitted to the journal.

Line 141: now lines 152-154 add the justification for performing the assessment at 7-10 cm dilation since they were women who were not given epidural anesthesia.

Lines 242-246: added.

Lines 270-271: added in limitations.

Round 2

Reviewer 1 Report

Thank you for the opportunity to read this revised manuscript. I feel that the authors have adequately addressed all concerns and that this manuscript is a valuable contribute to the literature.